# The Google matrix controls the stability of structured ecological and biological networks

Lewi Stone[1,2]

May's celebrated theoretical work of the 70's contradicted the established paradigm by demonstrating that complexity leads to instability in biological systems. Here May's random-matrix modelling approach is generalized to realistic large-scale webs of species interactions, be they structured by networks of competition, mutualism or both. Simple relationships are found to govern these otherwise intractable models, and control the parameter ranges for which biological systems are stable and feasible. Our analysis of model and real empirical networks is only achievable on introducing a simplifying Google-matrix reduction scheme, which in the process, yields a practical ecological eigenvalue stability index. These results provide an insight into how network topology, especially connectance, influences species stable coexistence. Constraints controlling feasibility (positive equilibrium populations) in these systems are found more restrictive than those controlling stability, helping explain the enigma of why many classes of feasible ecological models are nearly always stable.

[1] School of Mathematical and Geospatial Sciences, RMIT University, Melbourne, Victoria 3000, Australia. [2] Biomathematics Unit, Faculty of Life Sciences, Department of Zoology, Tel Aviv University, Ramat Aviv, Tel Aviv 69978, Israel. Correspondence and requests for materials should be addressed to L.S. (email: lewistone100@gmail.com).

Some of the outstanding unsolved challenges in theoretical biology concern the puzzling relationships between the feasibility, stability and complexity of biological systems[1–14]. Robert May's seminal paper[1] from the 70's, drew on the emergent patterns of ensembles of large random matrices to demonstrate that more complex and highly interconnected ecosystems are less likely to be stable, (that is, in terms of returning to equilibrium after disturbance). The approach has been applied widely in many other disciplines, ranging from systems biology, neurosciences, social network theory, to economics and banking systems[15,16]. May's analysis has been broadened in recent years as methods for analysing random matrices have advanced[2–4]. In addition to stability, any credible ecological model must maintain the basic constraint of feasibility, that all species present must have positive population abundances at equilibrium[5,6]. However, for systems of only minimal complexity, the study of feasibility becomes mathematically intractable. Fortunately, progress over the last decade in network science has made exciting new approaches available. From this viewpoint, Rohr, Saavedra and Bascompte (RSB) in ref. 6 introduced a framework which characterizes the range of parameters possible to simultaneously conserve feasibility and stability in large complex networks.

Here we propose a completely new direction based on a powerful reduction approach for studying complex systems having large-scale interaction architectures. Mutualistic pollination networks, for example, have blocks of competitive and mutualistic interactions[6,7], which often drown out the presence of other subtle underlying factors that might matter more, and can result in complex stability transitions. A basic understanding of these systems is still lacking[6,7]. By peeling away community-wide background interactions, simpler conditions for feasibility and stability can be derived. We make use of the same basic mechanism that sits at the heart of Brin and Page's[17] 'Google matrix' which ranks web pages, as it sifts through billions of hyperlinks across the entire world-wide-web[17,18]. The method provides a new way of working with time-honoured ecological interaction matrices. From this perspective, the Google matrix was made use of in Mathematical Biology (Stone[19] (1988)), some 10 years before it was invented by Google (Supplementary Note 4).

## Results

**Competition model.** The Lotka–Volterra (LV) equations of interspecific competition have been a source of tremendous inspiration for ecologists. The equations, for $n$-competing species, read[5,13,14,19–22]:

$$\frac{dN_i}{dt} = r_i N_i \left(1 - \sum_{j=1}^{n} a_{ij} N_j\right). \qquad (1)$$

Here $N_i$ is the abundance of the $i$'th species, while the positive parameter $r_i > 0$ defines its birth-rate. Central to our work is the interaction matrix $\mathbf{A} = (a_{ij})$. The competition coefficient $a_{ij} > 0$ measures the negative impact species-$j$ has directly on species-$i$. Intraspecific competition coefficients for individuals within a species are scaled to unity $a_{ii} = 1$, as are the carrying capacities of all species. These commonly adopted scalings have a long historical justification[1,5,13,14,19–22], but may also be relaxed (Supplementary Note 1K).

In the naive 'uniform competition model' each species competes with every other with equal strength $a_{ij} = c$, ($0 < c < 1$). To incorporate the vagaries of the real world, the limited uniform model may be 'brought to life' by incorporating stochasticity that acts to structurally disturb interaction parameters[14,19–23]. A large ensemble of competitive communities may be specified all of which on the average, resemble the uniform model:

$$a_{ij} = c + b_{ij}, \qquad (2)$$

with mean interaction strength $<a_{ij}> = c$. The structural disturbance matrix $\mathbf{B} = (b_{ij})$ has elements of mean zero, uniformly distributed in the interval $[-cv, +cv]$ with 'spread' $v$ ($0 \leq v \leq 1$), and variance $\mathrm{Var}(b_{ij}) = \sigma^2$. In this model, environmental fluctuations make the interaction strengths vary about the mean strength of competition $c$. Thus two communities may both have the same average strength of competition $c$, but the one undergoing stronger perturbations will show a greater variation in its interaction coefficients. Hence the stochastic model associates increasing disturbance with an increase in $\sigma^2$. We will find it convenient to represent the level of disturbance in the whole community by $\gamma$, where:

$$\gamma = \frac{\sqrt{n-1}\,\sigma}{1-c}. \qquad (3)$$

The ensemble contains the totality of possible interaction matrices. Each matrix is associated with its own equilibrium vector of population abundances. The subset of feasible solutions of (1), all have positive equilibria, with $N_i^* > 0$ for each species-$i$ ('*' indicating equilibrium). The feasible subset represents all possible candidates for survival as a persistent system.

For competition communities, the ecological interaction matrix $\mathbf{A}$ may be decomposed into three components: a background network of uniform all-to-all competitive interactions $\mathbf{C}$ ($c_{ij} = c$), a matrix of perturbations $\mathbf{B} = (b_{ij})$ and the self-regulatory interactions via the diagonal identity matrix $(1-c)\,\mathbf{I}$. Then $\mathbf{A} = (1-c)\mathbf{I} + \mathbf{B} + \mathbf{C}$, or in full:

$$\mathbf{A} = \begin{bmatrix} 1-c & b_{12} & . & . & b_{1,n-1} & b_{1,n} \\ b_{21} & 1-c & . & . & b_{2,n-1} & b_{2,n} \\ . & & & & & . \\ . & & & & & . \\ b_{n-1,1} & & & & 1-c & b_{n-1,n} \\ b_{n,1} & b_{n,2} & . & . & b_{n,n-1} & 1-c \end{bmatrix} + \begin{bmatrix} c & c & . & . & c & c \\ c & c & . & . & c & c \\ . & . & . & . & . & . \\ . & . & . & . & . & . \\ c & c & . & . & c & c \\ c & c & . & . & c & c \end{bmatrix} \qquad (4)$$

The first matrix on the Right Hand (RH) side is May's matrix of fluctuations $\mathbf{A_M} = (1-c)\mathbf{I} + \mathbf{B}$, while the second is the rank-one matrix $\mathbf{C}$.

**Stability of competition model.** Under what conditions are feasible model competition systems stable? Recall that local stability guarantees that an ecosystem will return to equilibrium after a 'small' population perturbation, while global stability ensures return to equilibrium for any sized population perturbation[1,24]. Theoretical ecologists study matrix eigenvalues ($\lambda_i$) of the stability matrix $\mathbf{S} = \mathbf{DA}$ to determine local stability [where diagonal matrix $\mathbf{D} = diag(N_i^*)$]. It is important to emphasize that for the models studied here, for feasible systems ($\mathbf{D} > 0$), the matrix $\mathbf{S}$ is locally stable if all the eigenvalues of the matrix $\mathbf{A_M}$ have positive real parts (Methods section).

A major achievement of May[1] was to characterize the stability properties of the random matrix of fluctuations $\mathbf{A_M} = (1-c)\mathbf{I} + \mathbf{B}$ when competition is absent ($\mathbf{C} = 0$). This reflects the stability of all those systems close to equilibrium in which interactions are equally likely to be positive or negative. May[1] demonstrated that a typical random community will be locally stable if the interaction disturbances are 'not too large,' namely:

$$\gamma < 1 \qquad (5)$$

and unstable otherwise (Fig.1a). The larger the number of species $n$, the sharper the transition from stability to instability at $\gamma = 1$ (Fig. 1a). However, the analysis does not give direct information

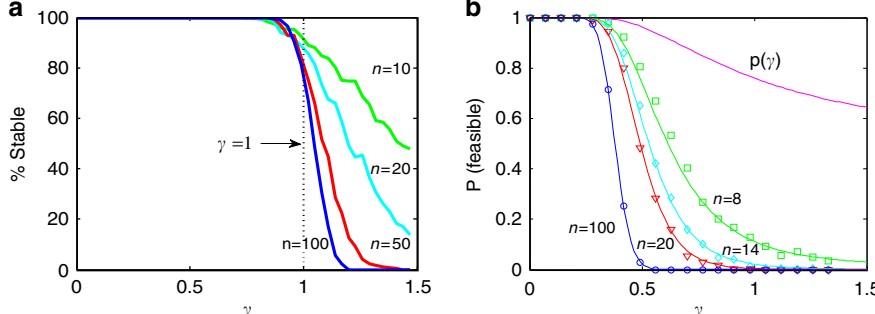

**Figure 1 | Local stability and feasibility. (a)** Percentage of locally stable 'May matrices' $\mathbf{A_M}$ as a function of disturbance $\gamma$ in an ensemble of 500 matrices for different community sizes $n = 10, 20, 50$ and 100. May's stability threshold sits at $\gamma = 1$. **(b)**. The probability of feasibility, $Pr(feasible)$, as a function of disturbance $\gamma$, for $n$-species competition with different community sizes $n = 1, 8, 14, 20$ and 100. Each probability marked by a square, circle and so on is the proportion of feasible systems in 500 runs of equation (1). Analytical prediction from Supplementary Note 1C displayed as continuous curves.

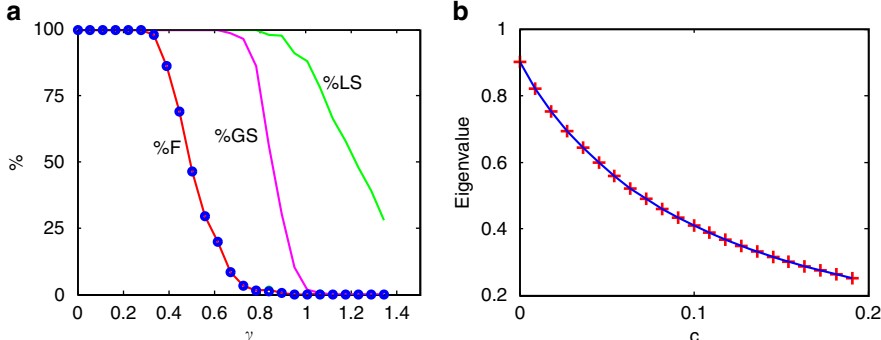

**Figure 2 | Competition model. (a)** Competition model. Characteristics of communities having $n = 20$ species, competition $c = 0.3$, as a function of disturbance $\gamma$. Percentage of 500 systems that were: (i) feasible: %F (red line); (ii) possessed locally stable interaction matrices $\mathbf{A}$: %LS (Locally Stable; green line); (iii) feasible together with locally stable interaction matrices $\mathbf{A}$: %F&LS (blue circles); (iv) having globally stable interaction matrices %GS (Globally Stable; magenta): (v) both feasible and having globally stable interaction matrices: %F&GS (also blue circles). The graphs indicate %F = %F&LS = %F&GS. **(b)** As the stability matrix $\mathbf{S} = \mathbf{DA}$ (blue line) is a Google matrix , its critical eigenvalue is identical to that of $\mathbf{S_M} = \mathbf{DA_M}$ (red +; see Methods-Google matrix). The critical eigenvalues of the two matrices lie exactly on the same curve when plotted as a function of competition strength $c$. Here the competition interaction matrix is $\mathbf{A} = \mathbf{I}(1 - c) + \mathbf{C} + \mathbf{B}$ and $\mathbf{A_M} = \mathbf{I}(1 - c) + \mathbf{B}$ with $n = 10$, $v = 0.4$.

about the stability of structured communities, such as communities of competition. Nevertheless, the matrix $\mathbf{A_M}$ will be shown to form the backbone of more complex ecological network models.

We begin by arguing that stability criterion (5) for the May matrix $\mathbf{A_M}$ proves to be the exact same stability condition for feasible LV-competition systems but for any $c$ $(0 < c < 1)$, in equation (1). To understand this, compare the matrices:

$$\mathbf{A} = \mathbf{I}(1 - c) + \mathbf{B} + \mathbf{C}; \quad \mathbf{A_M} = \mathbf{I}(1 - c) + \mathbf{B}. \quad (6)$$

In fact $\mathbf{A}$ is just $\mathbf{A_M}$, but perturbed by the uniform competition matrix $\mathbf{C}$. Quite remarkably, we show that the stability matrices associated with these two matrices are Google matrices (Methods section) and therefore have all eigenvalues, except for one, exactly the same (Fig. 2b; Supplementary Note 1F). The end result is that matrix $\mathbf{C}$, with all its many interactions, has little relevance for determining stability, which can be deduced solely from analysis of the reduced matrix $\mathbf{A_M}$ (Supplementary Note 1E). The result, which is not trivial, has gone unnoticed previously, but it is robust and holds almost exactly when the scaling of model (1) is relaxed (Supplementary Note 1K). The same underlying concept was taken advantage of in a different context to calculate PageRank via the Google matrix, in a way that takes into account the massive number of links across the entire world-wide-web[17] (Methods section). Putting this all together, we have found:

*Result A*. Feasible competition systems for any $c$ $(0 < c < 1)$, are locally stable if May's matrix $\mathbf{A_M}$ is locally stable, namely when $\gamma < 1$, and unstable otherwise. The larger the number of species, $n$, the sharper is the transition from stability to instability at $\gamma = 1$(Fig. 1a).

The conclusion should be seen as a restatement of May's[1] result for random communities, but now here widened to apply for random communities structured with competition. The parameter $\gamma$ representing the intensity of structural disturbances, completely governs stability in a simple manner. It should also be noted that we are chiefly interested in local stability, since fortuitously, feasible locally stable competition systems can be shown to be nearly always globally stable (Supplementary Note 1G; The exceptions are discussed in the Methods section.).

**Feasibility of competition model**. Following the lead from Roberts (1974, 1989), suppose we draw a system at random from the stochastic ensemble competition model, for fixed model parameters. We are interested in determining the probability of its feasibility, Pr(Feasible), for which all $n$-species have positive equilibrium populations. The probability Pr(Feasible) may be estimated as the percentage of feasible systems from a set of 500 random model systems as plotted in Fig. 1b. It is possible to show that for a given community size $n$, the probability is purely a function of $\gamma$ (Supplementary Note 1C). The model estimates of

Pr(Feasible) match analytical predictions closely (Methods section). Clearly the larger the number of species $n$, the more difficult it becomes to generate a feasible system. Moreover, Fig. 1b demonstrates that large feasible systems ($n > 14$) require that the variability of the structural disturbances is 'not too large,' specifically $\gamma < 1$.

Importantly, analytical and numerical analyses predict that almost all feasible competition systems sit in the range $\gamma < 1$ (Methods section, Supplementary Note 1H). Hence feasible systems must be stable (locally and globally; Methods section, Supplementary Note 1I) and certainly for large systems, all are stable ($n > 14$; Fig. 2a).

To understand this further, Fig. 2a displays the feasibility and stability characteristics of an $n = 20$ species community with $c = 0.3$, and plotted as a function of disturbance $\gamma$. These results were deduced numerically from equation 1. Almost all feasible model communities are locally stable, and nearly always globally stable. As $\gamma$ increases, the proportion of feasible systems (red) reduces to zero well before the proportion of locally (green) or globally (magenta) stable interaction matrices reduce to zero. As such, these calculations predict that nearly all feasible systems are stable, a characteristic that appears to be not restricted to the specific parameter ranges of Fig. 2a (Supplementary Note 1J).

However, this probability argument does not make transparent the key link between feasibility and stability. In Supplementary Note 1H a simple generic mathematical argument identifies this link, and can be explained via Fig. 3a. As shown, when $\gamma$ increases, at least one positive equilibrium population is destined to become unsustainably large in magnitude, and finally 'blows-up' at $\gamma \simeq 1$ when stability of the interaction matrix $\mathbf{A}$ is lost. Before 'blow up,' the large equilibrium values of some populations necessarily drive weaker species to extinction ($\gamma = 0.58$), and 'negative values,' explaining why feasibility is lost before stability of the interaction matrix $A$ is lost. This is the first theoretical prediction of the transition in a relatively general setting.

**Competition-mutualism (CM) networks.** The same methods can be extended to study more complex ecological systems, such as the animal-plant system presented in RSB (refs 6,7) in which mutualistic and competition networks operate simultaneously. Despite years of study, the stability properties of these systems remain poorly understood. Let $A_i$ and $P_i$ and denote the abundances of $n_1$ animal species and $n_2$ plant species.

Equations (1) then read:

$$\frac{dP_i}{dt} = r_i^{(P)} P_i \left( 1 - \sum_j c P_j + \sum_j m_{ij}^{(P)} A_j \right)$$

$$\frac{dA_i}{dt} = r_i^{(A)} A_i \left( 1 - \sum_j c A_j + \sum_j m_{ij}^{(A)} P_j \right)$$

(7)

Here, all plants species compete with each other with the same negative interaction strength $c$ ($0 < c < 1$) and likewise for animal species. Any interactions between plant species-$i$ and animal species-$j$ are mutualistic and positive ($m_{ij} \geq 0$).

The interaction matrix $\mathbf{A}$ may be split into its competitive and mutualistic blocks, and for the naive uniform model:

$$\mathbf{A} = \begin{bmatrix} 1 & c & -m & -m \\ c & 1 & -m & -m \\ -m & -m & 1 & c \\ -m & -m & c & 1 \end{bmatrix} = (1-c) \begin{bmatrix} 1 & 0 & 0 & 0 \\ 0 & 1 & 0 & 1 \\ 0 & 0 & 1 & 0 \\ 0 & 0 & 0 & 1 \end{bmatrix}$$

$$- \begin{bmatrix} 0 & 0 & m & m \\ 0 & 0 & m & m \\ m & m & 0 & 0 \\ m & m & 0 & 0 \end{bmatrix} + \begin{bmatrix} c & c & 0 & 0 \\ c & c & 0 & 0 \\ 0 & 0 & c & c \\ 0 & 0 & c & c \end{bmatrix} = (1-c)\mathbf{I} - \mathbf{M}^* + \mathbf{C}^*$$

(8)

The two diagonal blocks of matrix $\mathbf{C}^*$ represent the uniform competitive interactions within plants and within animals. Off-diagonal blocks of $\mathbf{C}^*$ are set zero given plants do not compete with animals for the same resources.

The two off-diagonal cooperative blocks define the matrix $\mathbf{M}^*$ of mutualistic interactions between plants and animals. Diagonal blocks are set to zero, since in this scheme animals (/plants) do not help their kind. Matrix $\mathbf{M}^*$, as shown schematically above, represents the uniform model of all-to-all interactions, although other network topologies are also explored. This includes connectance, whereby a proportion $(1-q)$ of randomly chosen nonzero interactions of $\mathbf{M}^*$ are set to zero, leaving a proportion $q$ nonzero.

Moving over to the stochastic ensemble model framework, nonzero mutualistic interactions are taken to be of the form $m_{ij} = m + b_{ij} > 0$, and thus all of the same average strength $m$. The mutualism matrix is now

$$\mathbf{M} = \mathbf{M}^* + \mathbf{B},$$

(9)

where $\mathbf{B} = (b_{ij})$ consists of random mean-zero structural disturbances, and where the uniform model corresponds to $\mathbf{B} = 0$. The $b_{ij}$ are uniformly distributed in the interval $[-mv, +mv]$ with 'spread' $v$ ($0 \leq v \leq 1$), having corresponding

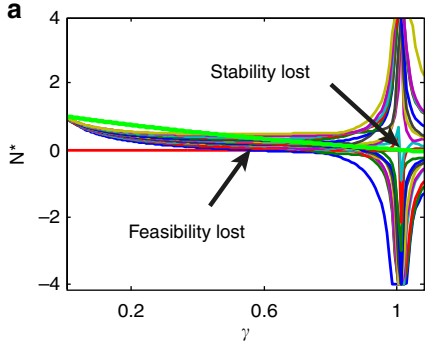
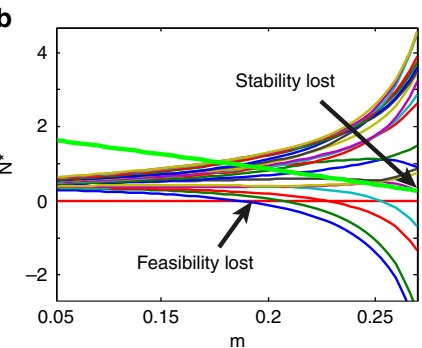

**Figure 3 | Equilibrium blow-up and loss of stability.** (**a**) Equilibrium abundances for competition model ($v = 1$, $c = 0.3$) with $n = 20$ species plotted versus disturbance level $\gamma$. The green line plots the real part of the critical eigenvalue of the interaction matrix $\mathbf{A}$ which zeroes when $\gamma = 1.04$, initiating instability. Feasibility is lost when $\gamma = 0.58$ and a population goes negative, well before stability of $\mathbf{A}$ is lost at $\gamma = 1.04$ where equilibria 'blow-up' see Methods section. (**b**) Equilibrium abundances of CM-model for $n = 20$ species ($n_1 = n_2 = 10$; $c = 0.2$, $q = 0.7$) plotted versus destabilizing interaction strength $m$. Feasibility is lost at $m = 0.19$. This is well before the critical eigenvalue (green line) zeroes at the equilibrium 'blow-up' point $m = 0.27$, where stability of $\mathbf{A}$ is lost.

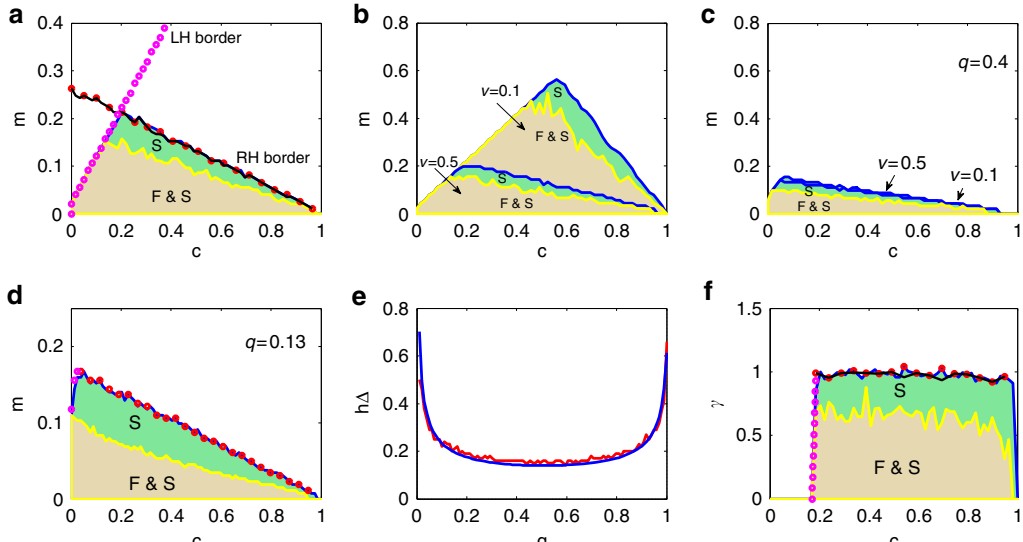

**Figure 4 | CM-model.** (**a**,**b**) with $n = 100$ species ($n_1 = n_2 = 50$), all-to-all mutualistic connectance $q = 1$ and disturbance spread $v = 0.5$. All systems are feasible and stable (F&S) in brown area of parameter space, which is here mutualistic strength $m$ versus competition $c$. Systems possessing stable interaction matrices (S) but not feasible are located in green area of parameter space. The 'Triangle of Stability' $\Delta$ has LH-stability border where $qm = c$ (magenta circles). The RH-Stability border is determined by the second-eigenvalue criterion equation (5) (red circles); and corroborated by numerical computations (blue line). (**b**) Similar to (**a**) with $q = 1$, but compares two disturbance levels, $v = 0.1$, and $v = 0.5$ by overlaying plots. (**c**) As (**b**) but with low connectance $q = 0.4$. (**d**) Canadian forest pollination matrix analysis $n_1 = 102$, $n_2 = 12$; $q = 0.13$. $v = 0.5$; (**e**) Height h($\Delta$) of triangle $\Delta$, versus $q$ (when $n_1 = n_2 = 50$; $v = 0.2$). Simulations (red) and mathematical prediction h($\Delta$) $\sim 1/\sqrt{nq(1-q)}$ (blue) from Supplementary Note 2D; (**f**) Testing May's criterion $\gamma_{CM} = 1$ (see below Result C). F and S plotted as function of disturbance $\gamma_{CM}$ versus $c$ ($q = 1$, $m = 0.2$) for $n_1 = n_2 = 50$ species.

variance $\text{Var}(b_{ij}) = \sigma^2$. We are chiefly interested in local stability of the interaction matrix **A** since fortuitously, feasible locally stable CM-systems are always found globally stable (Supplementary Note 2; RSB).

Consider then an $n$-species fully connected community ($n_1 = n_2 = n/2$ plants and animals; $q = 1$). Figure 4a outlines the feasibility and stability properties as a function of mutualism $m$ and competition $c$. The interaction matrix **A** is locally stable for all parameters inside the shaded prominent 'Triangle of Stability' $\Delta$ (green and brown). Note that the region where systems are both feasible and stable (brown), is necessarily enclosed within $\Delta$. Stability requires competition $c$ to lie in an interval between lower and upper bounds: strong enough to prevent runaway mutualism, but not too strong to promote competitive exclusion. Moreover the interval shrinks as mutualism increases in intensity, thereby creating the triangle's geometry. Figure 4b demonstrates that $\Delta$ can span large areas of parameter space, especially when disturbance is low (for example, $v = 0.1$), indicative of relatively high structural stability. The latter implies the existence of a relatively large area in parameter space where the model is both feasible and stable.

Contrast this pattern with a community of intermediate connectance $q = 0.4$ in Fig. 4c, where $\Delta$ proves to be relatively small in area. This small $\Delta$ holds for disturbance levels that are both low ($v = 0.1$) and high ($v = 0.9$), and is characteristic to a wide range of connectance levels $0.1 < q < 0.9$ as shown in Fig. 4e. Intriguingly, CM-systems are endowed with intrinsically poor structural stability for the intermediate connectances typical of real world networks (Fig. 4d). Note however, once connectance $q$ exceeds 50% structural stability actually increases with $q$; at first gradually but the increase is rapid once the interaction network becomes tight (Fig. 4e).

**Extending the Google matrix approach to CM networks.** Ideally we would like to be able to predict the stability and other characteristics of the CM-system from a knowledge of the empirical

interaction network **M**. It is not obvious that the Google matrix approach outlined above can contribute because the matrices involved are more complex in structure here. Quite remarkably though, the reduction approach can be successfully extended, and yields general criteria for mutualism matrices **M** of arbitrary topology.

Progress can be made by determining the borders of the triangle of stability $\Delta$ (Fig. 4a). The left-hand (LH) border is just the line $qm \simeq c$ (magenta circles; Supplementary Note 2D). On this border a species benefits from mutualism are on average equal to its losses from competition. Stability requires that mutualism levels be of limited intensity $qm < c$, which include all points in Fig. 4a below the LH-border.

The RH-stability border is found by applying the Google reduction to interaction matrix $\mathbf{A} = \mathbf{I}(1 - c) + \mathbf{C}^\star - \mathbf{M}$. The reduction makes it possible to discard the competition interaction matrix $\mathbf{C}^\star$, and leads to a simple condition based importantly, on $\lambda_2(\mathbf{M})$, the 'second-largest' or subdominant eigenvalue of the mutualism matrix **M** (Supplementary Note 2C). In Supplementary Note 2C it is shown that the RH-stability boundary in Fig. 4a corresponds to the line $\lambda_2(\mathbf{M}) = 1 - c$.

*Result B.* Summarising, stability requires that we consider only points in parameter space lying below the LH-border of the triangle $\Delta$. For these points: feasible CM-models are locally stable if

$$\lambda_2(\mathbf{M}) < 1 - c \qquad (10)$$

and unstable otherwise.

This establishes a direct connection between the topology of **M**, as coded into the eigenvalues[10] of **M**, and the stability of the CM-model. While many analyses (for example, ref. 10) focus on the dominant eigenvalue of **M**, this can lead to a wrong interpretation for understanding general stability. The above criterion (10) was tested on simulated model mutualism networks

in Fig. 4a,c,d,f and provides excellent predictions (red circles) of the true RH-stability border (blue).

The criterion (10) is appropriate whether or not $\mathbf{M}^\star$ has block structure or the disturbance matrix $\mathbf{B}$ is random, thereby opening the door for studying real empirical networks (Supplementary Note 3). As an example, consider the highly speciose mutualist-pollination network from Canadian spruce-fir forests[25], having $n_1 = 102$ insect species, and $n_2 = 12$ plants. The eigenvalue $\lambda_2(\mathbf{M})$ accurately identifies the RH-stability border of $\boldsymbol{\Delta}$ in Fig. 4d (red circles). The low connectance ($q = 0.13$) of the matrix effectively swivels $\boldsymbol{\Delta}$ to the left, and results in a mutualism-competition trade-off: intense mutualism can be maintained only for sufficiently weak levels of competition, and vice-versa. In this respect, the mutualistic network acts to reduce or minimize competition[7].

Note that the feasible systems (brown region) in Fig. 4 are contained fully within the triangle of stability $\boldsymbol{\Delta}$. Figure 3b makes clear a step-wise transition. As mutualistic strength ($m$) increase from zero, feasibility is lost first, followed by loss of stability of the interaction matrix $\mathbf{A}$ at higher levels of disturbance (Methods section; Supplementary Note 2E).

**All-to-all mutualism**. Returning now to the methodology, it is enlightening to take the Google reduction procedure one step further. Suppose that the matrix $\mathbf{M}^\star$ is significantly structured, for example, in all-to-all connected blocks with connectance $q = 1$. Now when investigating stability, both matrices $\mathbf{C}^\star$ and $\mathbf{M}^\star$ may be 'discarded' (Supplementary Note 2C). Stability may then be determined from the matrix: $\mathbf{A_M} = \mathbf{I}(1 - c) - \mathbf{B}$, which returns us back to the May criterion:

*Result C.* Assuming that we consider only points in parameter space lying below the LH-border (Supplementary Note 2E), then: feasible all-to-all block CM-systems are locally stable if matrix $\mathbf{A_M}$ is locally stable.and unstable otherwise (Supplementary Note 2D).

The criterion gives deeper insight into the determinants of stability in the CM-model. Namely stability is lost when the underlying stable uniform model is perturbed too severely. For the simplest case, where the number of animal equals the number of plant species ($n_1 = n_2 = \frac{n}{2}$, $q = 1$), the May stability condition requires $\gamma_{CM} < 1$, and instability when $\gamma_{CM} > 1$, where now $\gamma_{CM} = \sqrt{2n}\sigma/(1 - c)$ (Supplementary Note 2A). In Fig. 4a,f, the May criterion accurately predicts the border of stability at $\gamma_{CM} = 1$ (black line). The LH-border occurs as predicted at $m = c = 0.2$ (magenta).

**Real mutualism networks**. Although claims have been made that mutualism networks have strong internal structure[6], our analysis of 20 real empirical mutualism networks from RSB show them to have almost identical characteristics as their randomized matrix counterparts[26,27] in terms of two key parameters—their critical eigenvalue and species nestedness (Fig. 5; Supplementary Note 2F). Surprisingly, any internal topological structure in these networks cannot play a major role, assuming realistic biological constraints that preserve the network degree distribution[26,27]. The impact of these features on the eigenvalue appears to be minimal compared with the impact of connectance on structural stability, as just outlined.

## Discussion

In conclusion, while recent studies of the CM-model have failed to find any stability conditions or 'particular pattern in how the critical (stability) level of mutualistic strength varies with model parameters' (RSB), the techniques presented here result in strong clear relationships. Moreover, May's[1] early stability predictions equation (5) for large complex random systems and

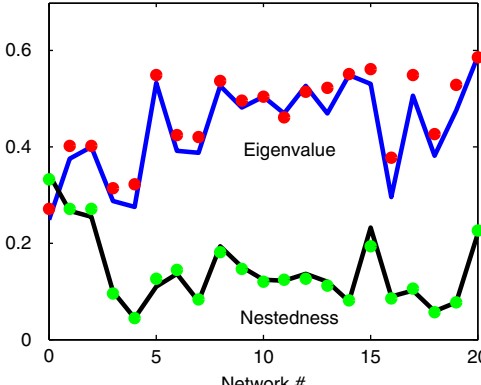

**Figure 5 | Nestedness.** The critical eigenvalue (blue line) of the interaction matrix and nestedness (black line; calculated as in ref. 26 for each of 21 empirical networks separated along x-axis (RSB; Supplementary Note 2F) where parameters taken as $c = 0.2$ and $m = 0.1$. Each red/green circle shows the average of the eigenvalue/nestedness of 25 randomized matrices[26,27].

equation (10), hold surprisingly well for competition, as well as highly networked CM-systems; local stability is lost when disturbances increase beyond a relatively small threshold level ($\gamma = 1$). This proneness to instability increases with the number of species, intensity of competition and level of disturbance. Analysis of structural stability, the range of parameter space for which feasibility and stability holds, leads to a different but not contradictory viewpoint, more in line with Elton[28]. Namely, CM-systems have poor structural stability for $0.1 < q < 0.9$, while tightly connected ecological networks have the highest structural stability. Ecosystem stability and vulnerability should be assessed by integrating the results from these two different frameworks.

The theory developed here also makes clear that constraints on feasibility are more restrictive than those on stability, and explains why nearly all feasible systems are stable for many classes of ecological models (Supplementary Note 2E). Interestingly, loss of feasibility might be viewed as an early warning precursor of the interaction matrix losing stability. Hence, external anomalies from changing climate, resource availability or environmental hazards, may readily lead to species extinctions often well before ecosystem instability can even be identified. The tools presented here, based on the Google matrix, extend the scope of May's study of large complex systems making it possible to untangle other important ecological interaction structures. These techniques can be readily adapted to a wide range of disciplines in network science.

## Methods

**Feasibility.** A calculation (Supplementary Note 1C, refs 19,20) shows that the probability a single arbitrary species has a positive equilibrium population is purely a function of $\gamma$, the variability of the structural disturbances, that is, $Pr(N_i^* > 0) = p(\gamma)$, as plotted in Fig. 1b (purple). A good approximation for the probability that all $n$-species are positive is then: $Pr(\text{Feasible}) = p(\gamma)^n$. The predictions of feasibility, Pr(Feasible) accurately match model simulations for a wide range of community sizes $n$ and parameters. This is seen in Fig. 1b which plots Pr(Feasible) as estimated as the percentage of feasible systems from a set of 500 random model systems. The figure shows that large feasible systems are only possible if the variability of the structural disturbances is 'not too large,' or in quantitative terms, if $\gamma < 1$ (cf equation (3)). Only then can $p(\gamma)$ be large enough to ensure that Pr(Feasible) is significantly greater than zero. Even for relatively small systems, the probability of feasibility is slight when $\gamma \approx 1$ (for example, Pr(Feasible) = 0.07 when $\gamma = 1$, $n = 8$).

**Stability.** The stability of equation (1) is determined by the eigenvalues of the stability or community matrix $\mathbf{S} = \mathbf{DA}$, where $\mathbf{D}$ *is the diagonal matrix* $\mathbf{D} = diag(N_i^*)$. Local stability is ensured iff all eigenvalues of $\mathbf{S}$ have positive real parts. For feasible competition systems, $\mathbf{D} > 0$, it is demonstrated in Supplementary

Note 1I that the matrix $S$ is locally stable iff all eigenvalues of $A_M$ have positive real parts. Global stability is ensured if the symmetric matrix $A + A^T > 0$ is positive definite (that is, has all eigenvalues positive; (24) and Supplementary Note 1G). In fact, nearly all feasible competition systems are globally stable, with rare counterexamples appearing only for $\gamma > 0.71$ when the number of species $n$ is not large (Fig. 2a (magenta)) as proven in Supplementary Note 1G.

**When Google meets Lotka–Volterra.** The simplest Google matrix is of the form:

$$G = (1 - c)A + cE$$

where $E$ is a matrix with $E_{ij} = 1/n$. The matrix $A$ is an $nxn$ stochastic matrix whose row sums add to unity, that is, $Ae = e$, $e = [1, 1, 1, …., 1, 1]'$. Matrix $A$ usually represents a scaled directed network such as the world-wide-web. The Google matrix $G$ has two special properties. (I) First, there is a left-eigenvector $\pi$ of $G$ such that $\pi G = \pi$, referred to as the PageRank vector, that provides a rank of the relative importance of the nodes (/webpages) in the network[17]. (II) A second property concerns the eigenvalues of $A$. Specifically, if $A$ has eigenvalues $\lambda_1 = 1, \lambda_2, …, \lambda_{n-1}, \lambda_n$, then the eigenvalues of $G$ are[18] $\lambda_1 = 1, (1-c)\lambda_2, …, (1-c)\lambda_{n-1}, (1-c)\lambda_n$. It is important that the 'damping factor' $c$ is in the range $0 < c < 1$, because this gauarantees a unique solution for the PageRank vector and one that can be computed in a fast way and whose convergence is assured.

A more general Google matrix is of the form $G = (1 - c)A + c\,uv^T$ where $A$ is an $nxn$ matrix (not necessarily stochastic) and $u$ is a right eigenvector of $A$. In the case of the Lotka–Volterra competition model, the interaction matrix is:

$$A = A_M + cE = (1 - c)[I + B'] + c\,e.e^T$$

where the matrix of ones is $E = e.e^T$ and. $e^T = [1, 1, 1, … 1, 1]$ and $A_M$ is the May stability matrix. But $A$ is not a Google matrix since $e$ is not a right eigenvector of $A$, and thus neither of the two properties above will hold. However, at equilibrium the LV equations also satisfy the additional constraint $AN^* = e$, where $N^*$ is the vector of equilibrium populations. Using this, it is easy to show that the stability matrix $S = DA$ is a Google matrix. Because of the scaling involved and property II above, all but one of the eigenvalues of $S = DA$ are identical to the eigenvalues of the May stability matrix $DA_M$ (ref. 19; Supplementary Notes 1F and 4).

**Feasibilty lost before stability.** For competition systems see main text. A similar generic mechanism is found for CM-systems. As the mutualistic interaction strength $m$ increases from zero in Fig. 3b, many of the equilibrium populations grow exponentially, and ultimately reach unsustainable levels. Thus feasibility is lost at $m = 0.19$, well before the population 'blow-up' point at $m = 0.27$ where stability of the interaction matrix is always lost. The mechanism reflects an intrinsic bifurcation instability of the CM-model whereby at high levels of mutualism, species with large abundances drive weaker species to low levels and then 'negative' values so that feasibility is lost before stability is lost. For this reason, all feasible CM-models examined here are stable, as explained in more depth in Supplementary Note 2E.

**CM-systems and the subdominant eigenvalue.** Intriguingly $\lambda_2(M)$ is a direct proxy for interaction variability and our analysis finds it composed of two components: $\lambda_2(M) = Rand_1 + Rand_2$ (Supplementary Note 2D). (i) $Rand_1$ represents the variability or 'spread' of the structural disturbances $b_{ij}$, via the parameter $v$. Surprisingly, this component has negligible impact unless connectance is extreme for example, $q \approx 1$. (ii) $Rand_2$ represents the randomness induced by connectivity itself, since each interaction has a probability $q$ of being nonzero. This component eclipses the former when $0.1 < q < 0.9$. Figure 4e shows how the height $h(\Delta)$ and thus area of the stability triangle, depends on $q$ according to both simulations and mathematical predictions (Supplementary Note 2D). Unusually, $h(\Delta)$ is almost constant and of low magnitude for $0.1 < q < 0.9$. In this regime, a network's connectivity has large restrictive impact on the area of $\Delta$, and thus structural stability.

**Data availability.** The study required analysis of 21 empirical mutualism networks. The data were chosen based on the networks analysed in the paper of Rohr *et al.* (ref. 6) and published at www.web-of-life.es.

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

## Acknowledgements

I am particularly grateful to Alan Roberts for sharing his deep insights into these research questions over many years. I thank David Ellison, Elad Shtilerman, Jinjian Wang and Jordi Bascompte for helpful discussions. The support of the Australian Research Commission grant DP150102472 is gratefully acknowledged.

## Author contributions

L.S. conceived and executed the project and wrote the paper.

## Additional information

**Competing financial interests:** The author declares no competing financial interests.

