## [Peer review file · Nature Communications]

REVIEWERS' COMMENTS:

Reviewer #1 (Remarks to the Author):

I think this paper by Lewi Stone is both exceptionally interesting and indeed important, and also is presented with admirable clarity and relevant detail. In the past, if my memory is correct, I almost always had at least a few comments for improving the manuscript which I reviewed. But in this case I find it very hard to suggest any improvements, because I think the paper as it stands is quite exceptional in presenting a clear and accurate analysis in a very complicated area. I tried hard to find some "helpful suggestions", if only to prove that I have read the paper carefully, but it seemed to me that it already was in excellent shape.

Indeed, I would go so far as to say that this is one of the most important manuscripts I have seen submitted to *Nature*, and this is no trivial statement. Given that it must have been something like a 100 papers submitted to *Nature* which I reviewed over the previous three or four decades.

I do fully realise that you would have preferred it if I had searched hard and found some constructive suggestions for improving the paper. I can assure you I did spend some time reading it carefully. But I think it is simply excellent, and it did seem to me a bit silly to be trying to manufacture comments in order to substantiate the claim that I have read the paper carefully (which I have).

I hope you may be reassured that the topics this paper addresses are in an area with which I am very, very familiar. And I think the paper is extremely important in very clearly addressing some important issues (admittedly, many of them raised by myself).

Again, I really do hope this rather unusual response is indeed helpful to you.

Reviewer #2 (Remarks to the Author):

This paper presents an important advance in theoretical ecology that will be of wide interest. It is also well-written and easy to read, which is likely to increase its impact substantially. The key point is a greatly improved understanding of the relationship between feasibility of multi-species communities (i.e., existence of multi-species equilibria) and their stability, including systems with competition and mutualism. The argument that explains why observed many-species communities are likely to be stable is elegant, compelling and likely to be highly cited.

The progress reported here has been possible because of the author's exploitation of the Google matrix approach in this novel context. Inferences made based on analytical work are confirmed using simulations, so the results can be considered extremely reliable.

Minor comment:

lines 60-62. The phrasing here is a bit confusing. Are you saying the method presented in this paper was already presented in your 1988 thesis? Or just that the Google matrix appeared in your thesis? Please clarify. Also, while it seems likely that there wasn't an earlier application of the Google matrix, it is possible in principle that there is another document somewhere with an appendix doing similar calculations. Perhaps rephrase as "making it an application of the Google matrix some ten years".

AUTHOR RESPONSE TO REVIEWERS' COMMENTS:

Reviewer #1 (Remarks to the Author):

I think this paper by Lewi Stone is both exceptionally interesting and indeed important, and also is presented with admirable clarity and relevant detail. In the past, if my memory is correct, I almost always had at least a few comments for improving the manuscript which I reviewed. But in this case I find it very hard to suggest any improvements, because I think the paper as it stands is quite exceptional in presenting a clear and accurate analysis in a very complicated area. I tried hard to find some "helpful suggestions", if only to prove that I have read the paper carefully, but it seemed to me that it already was in excellent shape. Indeed, I would go so far as to say that this is one of the most important manuscripts I have seen submitted to Nature, and this is no trivial statement. Given that it must have been something like a 100 papers submitted to Nature which I reviewed over the previous three or four decades. I do fully realise that you would have preferred it if I had searched hard and found some constructive suggestions for improving the paper. I can assure you I did spend some time reading it carefully. But I think it is simply excellent, and it did seem to me a bit silly to be trying to manufacture comments in order to substantiate the claim that I have read the paper carefully (which I have). I hope you may be reassured that the topics this paper addresses are in an area with which I am very, very familiar. And I think the paper is extremely important in very clearly addressing some important issues (admittedly, many of them raised by myself). Again, I really do hope this rather unusual response is indeed helpful to you.

Stone: No changes requested.

Reviewer #2 (Remarks to the Author):

This paper presents an important advance in theoretical ecology that will be of wide interest. It is also wellwritten and easy to read, which is likely to increase its impact substantially. The key point is a greatly improved understanding of the relationship between feasibility of multi-species communities (i.e., existence of multi-species equilibria) and their stability, including systems with competition and mutualism. The argument that explains why observed many-species communities are likely to be stable is elegant, compelling and likely to be highly cited. The progress reported here has been possible because of the author's exploitation of the Google matrix approach in this novel context. Inferences made based on analytical work are confirmed using simulations, so the results can be considered extremely reliable. Minor comment: lines 60-62. The phrasing here is a bit confusing. Are you saying the method presented in this paper was already presented in your 1988 thesis? Or just that the Google matrix appeared in your thesis? Please clarify.

These two things are tied together, because the method makes use of the Google matrix. I have clarified the text by being more specific and simplifying the language to read:.

The method provides a new way of working with time-honoured ecological interaction matrices. From this perspective, the Google matrix was made use of in Mathematical Biology [Stone19 (1988)], some ten years before it was invented by Google (Supplementary Note 4).

Also, while it seems likely that there wasn't an earlier application of the Google matrix, it is possible in principle that there is another document somewhere with an appendix doing similar calculations. Perhaps rephrase as "making it an application of the Google matrix some ten years".

Fixed. See above

This has been changed as requested.